



# The Land-Atmosphere Feedback Observatory: A New Observational Approach for Characterizing Land-Atmosphere Feedback

Florian Späth[1], Verena Rajtschan[2], Tobias K.D. Weber[2], Shehan Morandage[2], Diego Lange[1], Syed Saqlain Abbas[1], Andreas Behrendt[1], Joachim Ingwersen[2], Thilo Streck[2], Volker Wulfmeyer[1]

[1]University of Hohenheim, Institute of Physics and Meteorology, Stuttgart, Germany
[2]University of Hohenheim, Institute of Soil Science and Land Evaluation, Stuttgart, Germany

*Correspondence to*: Florian Späth (f.spaeth@uni-hohenheim.de)

**Abstract.** Important topics in Land-Atmosphere (L-A) feedback research are water and energy balances and heterogeneities of fluxes at the land-surface and in the ABL. To target these questions, the Land-Atmosphere Feedback Observatory (LAFO)
has been installed in Southwest Germany. The instrumentation allows comprehensive and high-resolution measurements from the bedrock to the lower free troposphere. Grouped in three components: atmosphere, soil and land-surface and vegetation, the LAFO observation strategy aims for simultaneous measurements in all three compartments. For that the LAFO sensor synergy contains lidar systems to measure the atmospheric key variables humidity, temperature and wind. At the land-surface eddy covariance stations are operated to record the energy distribution of radiation, sensible, latent and
ground heat fluxes. With a water and temperature sensor network the soil water content and temperature is monitored in the agricultural investigation area.

The observations in LAFO are organized in operational measurements and intensive observation periods (IOPs). Operational measurements aim for long timeseries dataset to investigate statistics as we present as example the correlation between mixing layer height and surface fluxes. The potential of IOPs is demonstrated with a 24 hour case study with dynamic and
thermodynamic profiles with lidar as well as a surface layer observation with the scanning differential absorption lidar to relate atmospheric humidity patterns to soil water structures.

Both long-term observations and IOPs are important for improving the representation of L-A feedbacks in climate and numerical weather prediction models.

## 1 Introduction

Land-atmosphere (L-A) feedbacks are the result of interacting processes related to the exchange of momentum, energy, and mass in the L-A system. This system consists of the compartments soil, land cover such as vegetation, and the lower troposphere, encompassing the atmospheric boundary layer (ABL) and the surface layer (SL), including their modifications due to human activities.

A profound understanding and representation of L-A feedbacks contributes to an improvement of the skill of both weather
(Holt et al. 2006) and medium-range to sub-seasonal forecasts (van den Hurk et al., 2012). L-A feedbacks have been



identified as one of the key science topics for advancing regional climate simulations (Jacob et al., 2020). L-A feedbacks influence the effects of historic, current, and future land use and land cover changes (LUCC) on regional hydrology, weather, and climate (Davin et al., 2020; Devanand et al., 2020; Stephens et al., 2021). It is of particular importance to understand these interactions over agricultural landscapes in order to maintain high crop yields and food security (Singh et al. 2018,

McDermid et al. 2019). L-A feedbacks over agricultural landscapes are also critically important because they modify the impacts of climate change over land (Seneviratne et al., 2006, 2010; Dirmeyer et al., 2012). In the evolution and strength of heatwaves and droughts (Jaeger and Seneviratne, 2011; Zhou et al. 2019a; Miralles et al., 2014, 2019) L-A feedbacks play a very important role. These extremes are expected to be sensitive to and amplified by climate change (Vogel et al., 2017, 2018; IPCC 2021) and can only be simulated correctly if the feedbacks and reactions of the vegetation due to water and heat

stress are well understood (Nolan et al., 2018; Anderegg et al., 2019; Zhou et al., 2019b). Advanced understanding of L-A feedback enables to consider bio-geoengineering efforts for mitigating the impacts of climate change (e.g., Davin et al., 2014; Branch and Wulfmeyer, 2019).

At any time and location, the quantities of the variables characterizing the L-A system such as the soil, canopy, and ABL temperatures and moistures as well as their fluxes and partitioning at the land surface are a result of these feedback processes

(Santanello et al., 2018; Wulfmeyer et al., 2018; Helbig et al., 2021). With a typical depth of ranging from several 100 m during night time to several km during the daytime, particularly the convective ABL (CBL) plays a very important role, as it rapidly responds (30-60 min) to changes of land surface properties by vertical mixing (Betts, 2004; Ek and Holtslag, 2004). For instance, van Heerwaarden et al. (2009) demonstrated that dry air entrainment in the CBL increases surface evaporation under all conditions. The strength of the L-A coupling depends on the incoming radiation, the land surface properties, and

the large-scale synoptic forcing. Therefore, improved understanding of L-A feedbacks requires resolving its diurnal cycle and comprehending how it depends upon large-scale conditions and the evolving land cover over a vegetation period. These interactions become particularly complex when clouds and precipitation develop, feeding back to soil moisture, vegetation photosynthesis and the surface energy balance (e.g., Betts et al., 2007; Gentine et al., 2013; Vilà-Guerau de Arellano et al., 2014; Chen and Dirmeyer, 2017).

To improve the understanding of the feedback processes, observations of energy and water fluxes in the soil and at the land surface are necessary. With variables to study L-A interactions at the surface are studied globally based on observation using eddy covariance (EC) stations. Many stations are organized in networks, e.g., FLUXNET (Baldocchi et al., 2001), AmeriFlux (Novick et al., 2018), ICOS (Franz et al., 2018), or TERENO (Zacharias et al., 2011) for which long-term dataset are available. Helbig et al. (2021) recently recommended to extent these observational stations with instruments for

atmospheric measurements, which will certainly facilitate better understanding of the feedback of the land surface with the ABL and the lower free troposphere.

In addition to ground-based EC stations, remote sensing systems from aircraft or satellites can also observe the land surface. Measurements of atmospheric variables with high temporal and spatial resolutions, however, are even more challenging. Fields of wind, temperature, and moisture must be provided at the same time, as these variables are entangled in all standard


SL schemes. Recently, a synergy of scanning lidar systems for wind, temperature and water-vapor became available fulfilling these measurement needs. For high-resolution line-of-sight wind measurements Doppler lidar (DL) systems use the coherent detection technique. The required range-resolved temperature measurements are realized with temperature rotational Raman lidar (TRL) (Hammann et al., 2015; Behrendt et al., 2015, 2020) and the necessary measurements of the moisture field are provided either from a water-vapor Raman lidar (WVRL) (Turner et al., 2002; Wulfmeyer et al., 2010) or

from a water-vapor differential absorption lidar (WVDIAL) (Wulfmeyer, 1999; Wagner et al., 2013; Wulfmeyer et al., 2016; Muppa et al., 2016; Späth et al., 2016). It was demonstrated that fluxes at the land surface and in the ABL can be measured. Tests of model parameterizations against lidar observations was executed by Milovac et al. (2016) and Muppa et al. (2016).

In 2017, the land-atmosphere feedback experiment (LAFE, Wulfmeyer et al., 2018) was, to the best of our knowledge, the first dedicated experiment to investigate L-A feedback with the deployment of a novel scanning lidar synergy. It took place

at the ARM Southern Great Plains site in Oklahoma, USA, and complemented the existing suit of instruments with a combination of Doppler lidars, Raman lidars, and differential absorption lidars to measure the wind velocity, humidity, and temperature from the land-surface through the ABL up to the lower troposphere (Späth et al., 2022; Lange et al., 2022). Operating scanning lidars were the key to obtain measurements in the surface layer region (Späth et al., 2022).

The campaign-based observations are time limited, thus, allow only few meteorological situations to be captured. For

statistically significant results and robust conclusions, long-term observations are required. For that reason, the Land-Atmosphere Feedback Observatory (LAFO) was set up as a research facility at the University of Hohenheim in Southwest Germany. The measurement data are comprehensive, highly resolved, and very precise, so that new parameterizations of land-atmosphere exchange processes between soil, vegetation and the lower troposphere can be developed, implemented, and tested. The design and operation of LAFO is connected to an international project of the World Climate Research

Program (WCRP) within the Global Land/Atmosphere System Study (GLASS) panel (see www.gewex.org/panels/global-landatmosphere-system-study-panel) named the Global Energy and Water Exchanges (GEWEX) LAFO (GLAFO, Wulfmeyer et al., 2020).

The overarching scientific goal of LAFO is to explore the L-A feedback with a novel synergy of energy balance/eddy covariance stations, vegetation measurements, and scanning lidar systems. LAFO is a platform to bring together existing and

unique instruments to seek for observations of the soil, the land surface, the vegetation and the lower troposphere simultaneously.

To achieve the overarching goal, LAFO follows three scientific objectives:

1. Determine the water and energy balances, and the land-atmosphere feedback as a function of the conditions of the soil, vegetation, and atmosphere in a study region with agricultural fields.

2. Investigate the heterogeneity of the fluxes at the land surface and in the boundary layer.

3. Develop new parameterizations of the fluxes at the land surface taking into account the vegetation dynamics and the turbulence in the ABL.





The LAFO measurement design follows the LAFE instrumentation and benefits from previous projects, e.g., the research

unit "Agricultural Landscapes under Global Climate Change – Processes and Feedbacks on a Regional Scale" of the German research foundation with long-term and on-going EC observations in Kraichgau and Swabian Jura area, southwestern Germany (Wizemann et al., 2015; Weber et al., 2022). This project contained also the Surface-Atmosphere-Boundary-Layer Exchange (SABLE) field campaign to test lidar observations in the surface layer (Späth et al., 2016). Both the LAFE and the new LAFO design with their instrument synergies have already made important contributions to this. Comparing

observations to models, e.g. the applicability of the Monin-Obukhov similarity theory (MOST) for natural heterogeneous land surfaces can be investigated or new model parameterizations can be developed (Lee et al, 2019, 2020; Späth et al., 2022).

This paper presents in Sect. 2 the observation strategy of the LAFO setup with the study area and its sensor synergy. The instrumentation is described in Sect. 3, followed by measurement examples in Sect. 4. Finally, we summarize and give an

outlook on future developments.

## 2 Observation Strategy

### 2.1 Observation Strategy

We set up a synergistic sensor network, to simultaneously monitor a wide variety of system properties and state variables from bedrock to lower free troposphere to accomplish our LAFO objectives. The sensor synergy consists of the three LAFO

components: 1) atmospheric measurements, 2) land-surface measurements, and 3) vegetation measurements (details of the components are given in Sect. 3). The interactions between the LAFO components are depicted in Figure 1and the measured variables are listed in Table 1. This suit of instrumentations illustrates the full extent of observations to study L-A feedbacks as proposed by the GLASS panel. The available and operated instruments in LAFO are described in Sect. 3.

The key atmospheric variables absolute humidity $q$, temperature $\theta$, and the turbulent fluctuations of the vertical wind $w$ are

captured with resolutions of 1-10 s in time and 30-60 m in space. The same holds for the by-products aerosol particle backscatter and extinction coefficient ($\beta_{par}$, $\alpha_{par}$) and their optical thickness ($\tau_{par}$), as well as the instantaneous boundary layer height $z_i(t)$. Furthermore, the horizontal wind profile $U(z)$ and potential properties like the atmospheric optical thickness $\tau_C$ can be retrieved. The combination of these variables delivers the following values: the dissipation of the turbulent kinetic energy (TKE) $\varepsilon$, profiles of the latent and sensible heat fluxes $L=<w'q'>$, $S=<w'\theta'>$ and the variances, $<q'^2>$, $<\theta'^2>$, and

$<w'^2>$, mixed terms of higher orders like $<w'q'^2>$ and the molecular destruction rate of humidity and temperature $\varepsilon_q$ and $\varepsilon_\theta$ (Wulfmeyer et al., 2016, 2022a) which are fundamental for budget analyses. Uncertainties of these variables are separated in atmospheric variance and instrumental noise with the autocovariance method described in Lenschow et al. (2000), Turner et al. (2014), and Wulfmeyer et al. (2016). The shear velocity $u^*$ and the surface latent and sensible heat fluxes $L_0$ and $S_0$, the





convective scales of vertical wind, humidity and temperature $w*$, $q*$, $\theta*$ are determined by combining the scanning lidars
with MOST (Späth et al., 2022) or with EC stations.

Vegetation variables and their spatial pattern in standing crops will be studied with multi- and hyperspectral remote sensing (unmanned aerial vehicles (UAV), phenotyping platform BreedVision, Busemeyer et al., 2013). This includes biomass, plant height $z_p$, plant density $\rho_P$, plant humidity content $q_P$, normalized difference vegetation index (NDVI), leaf area index (LAI) and nitrogen supply (red edge inflection point, REIP, camera mounted on UAV). The skin temperature $T_S$ of bare soil will be
measured with UAV.

The Water and Temperature Sensor Network (WaTSeN) monitors the spatial variability of soil water content $q(-z)$ and soil temperature $\theta(-z)$ in 40-cm depth. Additionally, the evaporation of the footprint can be derived from these data to identify energy balance errors of the EC stations (Ingwersen et al, 2015; Imukova et al., 2016). Profiles of soil water content and matric potential are measured at the EC stations. The variability of the soil properties can be studied with geophysical
measurements (e.g. EM38) and agricultural measurements (traction, Schutte, 2005) including soil probes.

The temporal dynamics and spatial configuration of the vegetation state is characterized routinely by LAI measurements (e.g., LICOR LAI 2000) and management data are provided by the field index cards of the agricultural plots. For the sub surface characterization of root growth dynamics, root density distribution, and root water uptake at various soil depths, additional field campaigns can be applied to complete the full picture.

With these data, the scientific objectives of LAFO can be achieved in the following way:

I. Determination of the water and energy balances and the LA feedback as a function of soil, vegetation, and atmospheric conditions in an agricultural study area. By combining the measurements of the LAFO components 1 and 2, all terms of the water vapor budget can be measured. Thus, the evolution of the boundary layer can be related to the state of the soil and the vegetation (LAFO component 3). This coupling of variables under different atmospheric conditions is sought
(e.g., Dirmeyer, 2014) to understand and predict changes in the components of the water budget, including clouds and precipitation due to internal and external factors (e.g., climate change) in the region. These data will also be used to help determine the accuracy of models in simulating the coupling.

II. Investigation of the heterogeneity of fluxes at the land surface and in the boundary layer: The usual neglect of the heterogeneity of the fluxes leads to errors in weather prediction and climate models. When these heterogeneities are
resolved, new scaling approaches can be developed and their impact on the coarse-scale used in models can be investigated. Furthermore, measuring the states of the land surface and boundary layer at the aimed high resolution will allow us to address the closure problem of EC measurements (Mauder et al., 2020; Ingwersen et al. 2015). A quantitative solution to the energy balance closure problem would enhance the confidence in the data from the international eddy covariance networks (e.g., FLUXNET) when being used for model validation.

III. Development of new parameterizations of the fluxes at the land surface under consideration of vegetation dynamics as well as turbulence in the atmospheric boundary layer: parameterizations of fluxes at the land surface require accurate data on soil and vegetation properties and states in combination with measurements of variables and fluxes in the atmospheric



soil layer. This complete set of LAFO measurements will also be used to further develop dynamic plant growth photosynthesis models (Gayler et al., 2013, 2014). In addition, research will also be conducted on how hyperspectral

measurements can be more accurately simulated using model forward operators and compared to model data.

For this purpose, we will use the WRF-NOAHMP model system, which includes plant dynamics components (GECROS model) and hydroecological components (HYDRO models). It can be used for both weather forecasts and climate simulations (Schwitalla and Wulfmeyer, 2014; Warrach-Sagi et al., 2013). The model already provides an ensemble of parameterizations, so that probabilistic investigations of their performance are also possible. Initial approaches to

combining lidar data to analyse turbulence parameterizations were presented in Milovac et al. (2016). By combining different types of measurements as envisaged in LAFO, parameterizations can be determined much better and more reliably (Wulfmeyer et al., 2016).

## 2.2 Study Area

LAFO site is located approximately 10 km south of Stuttgart city and 3km north from the Stuttgart airport, near the

University of Hohenheim, Stuttgart (48° 42' N, 9°11' E) in Southwestern Germany. The study site belongs to the university's agricultural research farm "Heidfeldhof" and comprises 26 agricultural fields covering a surface area of 84 ha. Figure 2 shows a map of the LAFO study area. The agricultural fields are surrounded by suburban areas with an adjoining forest in the westerly direction. The Heidfeldhof farm is owned by the University of Hohenheim and used as a research site for agricultural experiments. Main crops are maize, rapeseed, wheat, triticale, and barley cultivated in rotation, which include

other crops, too. The LAFO experimental site is flat; its elevation varies between 390 and 420 meters. The monthly temperatures vary between 0°C in winter and 18°C in summer months, and annual precipitation is 685 mm. An irrigation system is not installed. The soil type of the site is classified as Haplic Luvisol (periglacial loess) with a silty-loam texture (7 % sand, 73 % silt, and 21 % clay).

In addition to the LAFO measurements additional observations are made close by. In 2 km distance, the Hohenheim weather

and climate station has been recording data since 1878. The meteorological observations at the Stuttgart airport (METAR code: ESSD) is located only 3 km to the South. The official weather station Stuttgart-Schnarrenberg operated by the German weather service (DWD) with radiosonde launches (two per day) is located 13 km to the Northwest. Finally, the water level of the small Körsch creek next to the study area is recorded 8 km downstream to the East in Denkendorf.

## 2.3 Sensor Synergy

We have adopted a two-level observation strategy: long-term time series and intensive observations periods. These provide us with parsimonious yet highly informative data sets obtained from our sensor synergy network.

On the long-term, we operate a number of instruments in operational mode to obtain continuous year-round time series. As such, since the beginning of LAFO in 2018, WaTSeN has been continuously supplying time series to characterize the spatial heterogeneity of soil water contents and temperatures at the Heidfeldhof. Similarly, two EC stations have been deployed. At





the lidar site two Doppler lidars, a Doppler cloud radar, a micro rain radar and an optical distrometer have been continuously
operated since May 2020. During the vegetation period LAI measurements are taken from the fields with WaTSeN stations.
In addition to the operationally collected data, intensive observation periods will be organized as field experiments thereby
complementing the operational instrumentation by additional sensors and measurement systems to get an almost full picture
of the situation in the atmosphere, at the land surface, the vegetation and in the soil. This extended instrumentation is

operated temporarily to target a certain aspect of L-A interaction. Additional instruments of UHOH are DIAL and RRL,
BreedVision, Drones, EM38. Interesting sensors to add are further DLs for SL sampling, fiber-optical distributed sensing
(FODS) to study the temperature distribution, canopy sensors to observe temperature, humidity, radiation, and wind inside
and at the top of the canopy stand, and isotope measurements to discriminate between evaporation and transpiration.
Interested external partners can benefit from our sensor synergy and join through collaboration. For example, LAFO was one

observation site in the frame of the field campaign Swabian MOSES which took place in spring 2021 (Kunz et al., 2022).
During this field experiment the Atmospheric Raman Temperature and Humidity Sounder (ARTHUS) was operated to
capture atmospheric temperature and humidity profiles next to wind and cloud observations with DL and a Doppler cloud
radar.

## 3 Three LAFO components

The three LAFO components of the LAFO synergy are dedicated to the three compartments atmosphere, soil and land-
surface, and vegetation and are described in the following.

### 3.1 First LAFO component: Atmospheric measurements

The first component consists of three scanning lidar systems, a water vapor differential absorption lidar (DIAL), a water
vapor and temperature rotational Raman lidar (RRL), and Doppler lidars (DLs) for wind measurements which are located at

the lidar site (Figure 2). The DIAL and RRL have been developed and designed at the Institute of Physics and Meteorology
of UHOH and the deployed Doppler lidars are manufactured by Halo Photonics Ltd. (UK). With this combination, it is
possible to measure 2D to 3D structures of these important atmospheric variables from the ground through the boundary
layer to the lower free troposphere (about 3-4 km altitude). The DIAL is mounted on a mobile platform with a frequency-
doubled Nd:YAG-pumed Ti:sapphire laser transmitter at 818 nm with an output power of up to 10 W (Wagner et al. 2013;

Metzendorf, 2018). The 3D scanner is equipped with an 80 cm primary mirror and allows scanning of the full upper
hemisphere enabled for the first 3D humidity measurements (Späth et al., 2016). The RRL is also a mobile, 3D scanning
lidar system. The laser transmitter is based on a frequency tripled Nd:YAG laser at the eye-safe wavelength of 355 nm and a
40 cm receiving telescope with a 2-mirror 3D scanner (Radlach et al., 2008). A very efficient receiver separates the
components of rotational Raman scattering for temperature and vibrational Raman scattering for water vapor mixing ratio

(WVMR) measurements (Hammann et al., 2015). Meanwhile, a more compact and automated Raman lidar for temperature





and humidity measurements (Atmospheric Raman Temperature and HUmidity Sounder – ARTHUS, Lange et al., 2019) has been developed and is available at LAFO for collecting vertical profiles of temperature and humidity. Wulfmeyer et al. (2015) demonstrated that the DIAL and the RRL are currently the most accurate and highest resolution water vapor and temperature remote sensing systems in the world. The RRL was the first to measure inversions at the top of the boundary

layer and turbulent fluctuations in daytime temperature (Behrendt et al., 2015).

The humidity and temperature measurements are complemented by wind measurements of two DLs. One DL is dedicated to observe the vertical wind with turbulent resolving resolution of 1 s and 30 m in a constant vertical steering mode. The other DL is operated in 6-beam VAD (velocity azimuth display) mode to detect profiles of horizontal wind (Choukulkar et al, 2017). These measurements also have turbulence resolution. Wulfmeyer et al. (2016) showed that this measurement

combination of DIAL, RRL, and DL is crucial for measuring fluxes through the ABL and for developing new turbulence parameterizations. In combination with additional cross-track scanning DLs the measurement setup for SL profiling (Späth et al., 2016) can be achieved and used for mapping land surface fluxes. Additional instruments, e.g., DLs for cross-track scanning will be provided by project partners.

During cloudy and rainy weather situation one Doppler cloud radar (DCR) is operated in vertical stare mode. The DCR is a

MIRA-36 of Metek GmbH (Görsdorf et al., 2015). To characterize rain in more detail a micro rain radar (MRR, Metek GmbH) and an optical distrometer (ParSiVel) are installed as well. All instruments except DIAL and RRL are operated continuously and quicklooks of measurements are available in near real-time on the LAFO website https://lafo.uni-hohenheim.de/en.

### 3.2 Second LAFO component: soil and land-surface measurements

The second component combines sensors that determine energy fluxes at the land surface as well as state variables of soil.

Surface energy fluxes are derived from long-term eddy-covariance measurements at two stations at the Heidfeldhof (Figure 2). The stations are each fully equipped with a 3 m mast and sensors to measure $CO_2/H_2O$ concentrations, 3D wind components with a sonic anemometer, and up-/downwelling shortwave (SW) and longwave (LW) radiation to access the net radiation $Q_n$. This instrumentation allows to investigate the energy partitioning of $Q_n$ into sensible and latent heat fluxes $S$

and $L$, respectively, as well as ground heat flux $G$. All meteorological variables like air temperature, humidity, pressure and precipitation are collected, as well as soil water content, matric potential $h(-z)$, and soil temperature at five soil depths. The used sensors, characteristics, and methods are identical to Wizemann et al. (2015).

Beginning in June 2018 a Water and Temperature Sensor Network (WaTSeN) was installed to quantify the spatial-temporal patterns of precipitation, soil water contents, and temperature with a total of 22 Aquaflex II universal sensors (Streat

Instruments Ltd) to monitor the soil water content and temperature and 22 rain gauges (Pronamic) in November 2020, covering the entire Heidfeldhof (HFH; see Figure 2). Aquaflex II sensors are installed below the ploughing depth in 40 cm depth providing year-round measurements without laborious sensor deinstallation. Rain gauges, solar panels, and loggers are mounted on a post next to each field. Stations are named according to site position and field site number with the prefix





"HFH". The stations are powered by solar electric panels. Each station is equipped with Adcon Remote Transmission Unit
(RTU) data logger (OTT HydroMet GmbH) for automatically collecting and transmitting data via a 3G/4G network. Data
loggers store sensor data taking the mean over at 15 minute time intervals and data are then sent to the central database every
six hours. Since factory calibrated volumetric water contents of Aquaflex II sensors are highly biased (Poltoradnev et al.,
2015), the sensor signals are recalibrated based on site-specific soil sampling analysis. For this, soil core samples are
systematically collected four times per year in close proximity to each soil sensor. The samples are used to determine the
gravimetric water content by oven drying and converting them to volumetric water content by the determined bulk density.
The determined values subsequently used for calibration of the Aquaflex II sensor data.

In Figure 3 the spatio-temporal variability of soil water content (SWC), its standard deviation and time series of precipitation
for the two-year period 2019-2020 is shown. The SWC shows the typical wetting and drying cycles during the year reaching
field capacity in winter with long periods of drying between April and October. From the state-space of the standard
deviation of SWC ($\sigma_{SWC}$) in Figure 3, we conclude that the variation in SWC is controlled by the land cover (through
interception and absorption of plants and evaporation of soil) and by the fact that the infiltration fronts at 40 cm depth is far
less homogenous ($\sigma_{SWC}$ higher) than drying ($\sigma_{SWC}$ lower).

So far, we have not detected strong coupling between the soil hydrological variables and fluxes at the land surface during the
growing seasons. Latent and sensible heat fluxes show no significant correlation with SWC or water potential any depths.
Thus far, soil variables have not been shown to control the fluxes at the land surface during the growing season, and heat
fluxes are radiation driven. This can be beneficial since it allows to identify the plant state as the principal control for spatial
surface flux heterogeneity at the land surface.

## 3.3 Third LAFO component: vegetation measurements

This component consists of equipment for characterizing vegetation. These include UAVs for recording plant characteristics,
which now represent an innovative alternative to traditional remote sensing with manned aircraft or satellites due to the
inexpensive and short-term availability of high-spatial-resolution sensor data. Especially the higher spatial resolution allows
applications on smaller field plots. For the determination of vegetation indices for the derivation of biomass and nitrogen
supply, several spectral bands are necessary (e.g. 670, 700, 740 and 780 nm) for the calculation of the Red Edge Inflection
Point (REIP). The spectral analysis was performed with a UAV equipped with converted industrial 4-sensor camera for
REIP imaging by means of suitable interference filters (Geipel et al., 2014). In addition, the camera has been coupled to an
external light meter to dynamically adjust the exposure time to the solar irradiance. The camera is individually
programmable and also allows image processing in near real-time.

In cooperation with the Osnabrück University of Applied Sciences, the State Seed Breeding Institute (LSA) at the UHOH
has developed the "BreedVision" phenotyping platform (Busemeyer et al., 2013). It consists of a carrier vehicle and a sensor
module. The carrier vehicle is a high-wheel tractor especially adapted to the requirements of the sensor module. Both the
track width and the height of the carrier vehicle can be adjusted hydraulically. The sensor module includes sensors with





different morphological or spectral selectivity such as light grids, laser distance sensors, multi-reflective ultrasonic sensors, digital cameras, a plenoptic camera, and a hyperspectral camera. For example, light grids create a shadow image from the transmitted light through the plants of the plots, from which plant height $z_p$ and density can be indirectly determined, while a
hyperspectral camera can be used to determine plant moisture content or parameters such as the normalized differential vegetation index (NDVI). The sensor module is continuously adjustable in height, allowing a constant distance of the sensors to the plants, even with varying plant heights. The particularly innovative feature of the developed system is the combination of sensor data for trait prediction. The spatial and temporal allocation of all sensor data to a single position and thus to individual plant organs in the crop, which is necessary for this, is so far unique in field phenotyping. In addition to its use in
plant breeding, the system can also be used to record, by means of the various complementary sensors, a range of trait parameters along a plant stand.

Further, we routinely register the leaf-area index (LAI), the crop height and phenomenological growth stage code BBCH at our WaTSeN stations throughout the main vegetation period. Furthermore, we have a set of canopy sensors to investigate temperature and humidity at different levels inside and right above the canopy top.

**4 Measurement examples**

The LAFO equipment and its data products are listed in Table 1. The list contains the currently operationally available instruments and measurement data but can be extended in the future or for measurement campaigns with collaborators complementing instrumentation for certain time periods.

**4.1 Dynamic and thermodynamic profiling for turbulence analysis**

For studying dynamics and thermodynamics in the ABL at LAFO we use the synergy of DLs and ARTHUS. As explained in Sect. 2.3, we measure dynamic profiles of vertical wind with one constantly vertical pointing DL and retrieve the horizontal wind from LD in 6-beam VAD mode. In Figure 4 we show 24-hour wind measurements for 27 June 2021. The figure shows in (A)-(D) vertical profiles of vertical and horizontal wind as well as profiles of the backscatter coefficient. The resolution of the vertical wind and backscatter coefficient is 1 s and 30 m. The horizontal wind is retrieved from the 6-beam VAD mode
with a sinusoidal fit and results in a 90 s and 21 m temporal and spatial resolution (Bonin et al., 2017). To reduce the effect of the convective eddies for the horizontal wind, an additional gliding average of 21 minutes was applied. In the vertical wind plot (A) the development of the CBL started around 07 UTC which is indicated by the stronger up and down draft starting to grow from the surface into the boundary layer. The CBL stayed convective until around 17 UTC and reached a height of around 1 km. Before and after that period only low vertical wind was observed which is typical for the night time
boundary layer. While there was nearly no vertical motion at the beginning of the day a low-level jet (LLJ) in the horizontal wind up to 8 ms$^{-1}$ was recorded at a height of 500 m. This LLJ disappeared at around 10 UTC when the convective motion reached the height of the LLJ. Recording the data with turbulence resolution allows the derivation of further products like





vertical wind variance, momentum flux and turbulent kinetic energy which are plotted in (E)-(G). For these retrievals we determine the fluctuations to calculate the variance and higher-order moments (not shown here) following the method of

Lenschow et al. (2000) and Wulfmeyer et al. (2016). This method also allows to distinguish between atmospheric variance and instrumental noise contribution. The calculation of momentum flux and TKE follows Bonin et al. (2017). In the evening at 20 and 23 UTC two rain events were captured which are identified by reduced measurements range or strong downward motion and corresponding effects in the derived products.

The thermodynamic profiling was performed with ARTHUS. The time series of WVMR and temperature profiles of 27 June

2021 are shown in Figure 5 (A) and (B). The data show the moistening and warming of boundary layer in the cause of the day when the boundary layer became convective. In combination with vertical wind measurements the latent and sensible heat fluxes through the boundary layer can be calculated with the covariances of humidity and temperature fluctuations and vertical wind (Behrendt et al., 2020) and are plotted in Figure 5 (C) and (D). When including horizontal wind, the water vapor budget in the BL can be determined as demonstrated for the first time for a LAFE case by Lange et al. (2022).

**4.2 Correlation between turbulent mixing height and surface fluxes**

With the wind observation the turbulent motion in the BL can be captured. The depth of the CBL is related to the energy intake by solar radiation and its partitioning at the land surface. Thus, we determined the mixing height for the three-month period from 01 May to 31 July 2021 and correlate them to the surface fluxes measured with the EC station.

For determining the mixing layer height (MLH) we use vertically pointing Doppler lidar and Doppler cloud radar data and a

fuzzy logic approach to weight the different data (Bonin et al., 2018). The DL measures the vertical wind $w$, signal-to-noise ratio (SNR) and backscatter coefficient $\beta$ and from the DCR vertical profiles of radar reflectivity, Doppler velocity and spectral width are included in the algorithm. The main contribution comes from the vertical wind variance to map variance as turbulent for $<w'^2> \leq 0.1$ m²s⁻² or non-turbulent. A SNR threshold was set to filter the wind data for poor data quality and high $\beta$ values are used as indicator for the presence of clouds and precipitation. Clouds show high $\beta$ values and a strong $\beta$

gradients at the cloud bottom and precipitation is associated with high $\beta$ values and negative vertical wind. In addition, the DCR data are used to support cloud and precipitation estimates. Data of clouds and precipitation are filtered out and are not included in the determination of MLH.

The MLH is calculated for each day for the time period between 06:00 and 18:00 UTC which covers daytime between sunrise and sunset when turbulent mixing is expected. The analysis is based on the 1 s data. Nighttime MLH usually is very

low and mixing is caused by friction between air and land surface. These situations cannot be captured with our vertical pointing instruments as the lowest range bin is not lower than 100 m. Statistics of the 1 s MLH data are evaluated for 30-minute intervals as the fluxes are calculated likewise for half-hour timeslots. In Figure 6 (A) the statistics of the diurnal cycle of the MLH is plotted as whisker box plot. The box and whiskers provide a clear picture of data distribution. The box itself represents the range of 50 % central data (between 25 % and 75 %) also known as interquartile range (IQR). In the box the

red line gives the median of the data. The dashed lines (whiskers) extending from the box marking the remaining data





ranging from the minimum to the maximum values. Data points outside of these ranges are considered as outliers which is defined as < 1.5 IQR or > 1.5 IQR. The dashed curve and red dots follow the mean MLH. For the surface fluxes of sensible and latent heat and of the net radiation for the 30-minutes intervals also the whisker box plots are shown in Figure 6 (B)-(D). The correlation of the mean MLH and the fluxes was plotted in Figure 7.

The statistics include the full three-month period covering all cases including clear sky, clouds, rain, and thunderstorms. On convective days the MLH reached in the mean up to around 1.2 km and over all up to more than 2 km, however, on cloudy days the MLH stays low. This is also represented in the flux data when on cloudy days the surface heating is low. The sensible heat flux varies from 20 to 70 $Wm^{-2}$, latent heat flux varies from 20 to 300 $Wm^{-2}$, and net radiation from 20 to 400 $Wm^{-2}$. The growing of the MLH starts in the morning when convection starts with the surface heating after sunrise and

decays when the energy input decreases. Thus, MLH development is clearly related to the surface fluxes. The maximum is reached around noon when most energy is available. This is also reflected by the positive correlation of the mean MLH and the surface fluxes plotted in Figure 7. The MLH is directly correlated with sensible heat flux and net radiation whereas the correlation to latent heat flux shows a hysteresis effect. The growing MLH follows the increasing latent heat flux but the MLH decays in the afternoon even when the latent heat flux is still present and decreases not with a similar rate.

**4.3 Scanning lidar observations**

To observe the direct link between atmosphere and land surface we make use of the scanning capability of our lidar instruments. With low elevation scanning lidar measurements we are able to observe surface layer profiles of horizontal wind, humidity content, and temperature (Späth et al., 2022). Figure 8 demonstrates humidity measurements with the scanning WV DIAL across the agricultural fields to the northwestern direction. The scan direction and a photograph are

shown in Figure 8 (A). For the measurement the lidar scanning unit (Figure 8 (A)) moved with $0.2°s^{-1}$ for a 90° angle range and the data were analysed with 10 s and 67.5 m temporal and spatial resolutions, respectively. The measurement was performed on 20 October 2020 at 14:16 UTC. The plot in Figure 8 (B) shows four layers in the lowest 0.5 km of which the highest WV content is found in the lowest layer close to the ground and decreasing humidity content in the layers above. The lowest layer shows spatial variations along the scan axis with a lower humidity content at 500 to 600 m and an increase at

650 m. It is noteworthy that the high resolution of the WV DIAL measurement to be as sensitive as to resolve variabilities of less than 1 $gm^{-3}$. As the atmosphere is moistening from the surface, we can relate the structures close to the surface to land-surface properties as explained by Späth et al. (2016). Being already in autumn and most of the fields were harvested we make use of the WaTSeN stations. Here we measure the water content of the soil as indication for the potential of evaporation and transpiration at the land-surface. In Figure 8 (C) the soil water content of the WaTSeN stations HFH 1-9 is

given with respect to the distance to the WVDIAL. The course of the soil water content in Figure 8 (C) shows a minimum at 550 m distance and a maximum of water content at 700 m. This agrees with the behaviour of the atmospheric humidity observed with the lidar in the atmosphere above.



In the region with low atmospheric humidity below was intertillage on the fields in a small growing stage. Closer and farer the fields were bare soil where higher evaporation is enabled according to higher soil water content.

## 5 Summary and Outlook

Important topics in L-A feedback research are water and energy balances and heterogeneities of fluxes at the land-surface and in the ABL. To target these questions, LAFO has been installed as first GLAFO. LAFO is dedicated for L-A feedback research located in Southwest Germany in mid-latitude continental climate and demonstrates the GLASS panel proposed instrumentation allowing comprehensive and high-resolution measurements from the bedrock to the lower free troposphere. Grouped in three components: atmosphere, soil and land-surface and vegetation, the LAFO observation strategy aims for simultaneous measurements in all three compartments. For that, the LAFO sensor synergy contains lidar systems to measure the atmospheric key variables humidity, temperature and wind. At the land-surface EC station operated to record the energy distribution of radiation, sensible, latent and ground heat fluxes. With WaTSeN the soil water and temperature are monitored in the agricultural investigation area.

The observations in LAFO are organized in operational measurements and intensive observations periods (IOPs). Operational measurements aim for long time-series dataset to investigate statistics like demonstrated for the correlation between MLH and surface fluxes. Furthermore, the long-term dataset from the EC stations are interesting to be used for machine learning approaches to investigate new SL relationships as demonstrated by Wulfmeyer et al. (2022b). During IOPs non-automated instruments complements the operational instrumentation for extended analysis like analysis of sensible and latent heat fluxes in the ABL with humidity and temperature measurements from ARTHUS or SL observations of humidity with the scanning capability of the WV DIAL to relate atmospheric moisture distribution to soil water structures.

Further, some of LAFO's measurement technology has significant commercialization potential. In particular, the development of compact and operational water vapor and temperature profilers (ARTHUS, Lange et al., 2019; NCAR DIAL, Spuler et al., 2015) with high resolution and accuracy has long been requested for use in national and international networks to improve weather forecasting (Adam et al., 2016; Weckwerth et al., 2016; Thundathil et al., 2020, 2021), climate monitoring, verification of models and satellite measurements, and data assimilation. LAFO can serve as a platform to test and develop new devices of this kind.

In the upcoming months and years, not only will the observational dataset continuously grow but also be made publicly available. Out database is connected to the European Network of Hydrological Observatories (enoha, https://enoha.eu/) and allows access to and visualization of quality controlled timeseries of EC station data and of the WaTSeN data. Later also 2D dataset from the DLs and the DCR will be accessible. Until then, near real-time time-height cross-section plots of DL and DCR data are available on the LAFO website (http://lafo.uni-hohenheim.de/en). In addition to the observations, model runs of large eddy simulation will be prepared and performed with different configurations of the Weather Research and Forecast





(WRF) model (Bauer et al., 2022). With respect to model efforts, LAFO data can be used for model evaluation, process
studies or data assimilation ideas.

## 6 Data availability

The presented data will be made publicly available with acceptance of the manuscript. Until then, data are available on request to the corresponding authors. All raw data can be provided by the corresponding authors upon request.

## 7 Author contribution

VW, AB and TS designed the observation strategy and FS, VR, TKDW, SM, JI, DL carried them out. FS, VR, TKDW, SM, DL, and SSA analysed the data. FS, VR, TKDW prepared the manuscript with contributions from all co-authors

## 8 Competing interests

The authors declare that they have no conflict of interest.

## 9 Acknowledgment

We thank the Carl Zeiss Foundation for funding the setup phase of LAFO as part of its program to strengthen research infrastructures at universities. Further, we kindly thank Herbert Stelz, Stefan Pilz, Timo Keller, Thomas Schreiber, Christian Schade and Jacky Schulz for their different levels of support and measurements.

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





**Table 1** Measured variables with the LAFO sensor synergy.

| Instrument, Mode | Soil | Vegetation | Atmosphere | Fluxes | Radiation | Aerosols | Clouds |
|---|---|---|---|---|---|---|---|
| DIAL, vert | | | $q(z)$, d $q(z)/dz$, $q'(z)$, $<q'^2>$, $<q'^3>$ | | | $\beta_{par}(z)$ | |
| DIAL, RHI | | | 2D $q$, d$q/dz$ | | | 2D $\beta_{par}(z)$-field | 2D field |
| RRL, vert. | | | $T(z)$, d$T(z)/dz$, $T'(z)$, $<T'^2>$, $<T'^3>$ | | | $\beta_{par}(z)$, $\alpha_{par}(z)$ | Cloud bottom height |
| RRL, RHI | | | 2D $T$, d$T/dz$ | | | 2D $\beta_{par}(z)$, $\alpha_{par}(z)$ field | 2D field |
| DL, vert. | | | $w(z)$, $w'(z)$, $<w'^2>$, $<w'^3>$, $\varepsilon$ | | | $\beta_{par}(z)$ | Cloud bottom height |
| DL; VAD scan | | | $V(z)$, d$V/dz$ | | | | |
| DL, RHI | | | 2D wind field | | | 2D $\beta_{par}(z)$-field | 2D field |
| DIAL-DL, vert. | | | | | | | |
| RRL-DL, vert. | | | | | | | |
| Two DLs, crosstrack scanning | | | $u$, $v$ at the cross points | $u*$ at the cross point | | | |
| DIAL-RRL-DL, RHI | | | $w*$, $q*$, $T*$ | $u*$, $L(z)$, $S(z)$ | | | |
| EC station | $T(-z)$, $q(-z)$, $h(-z)$ | | $T$, $q$, $u$, $v$, $w$ (all in 2m) | $u*$, $L_0$, $S_0$, $G$, $Q_n$, precipitation | net SW and LW | | |
| UAV, Breedvision | $T_S$ | NDVI, LAI, $T_C$, $z_P$, $\rho_P$, $q_P$ | | | vis and IR spectra | | |
| WaTSeN | $T(-z)$, $q(-z)$ | | | | | | |


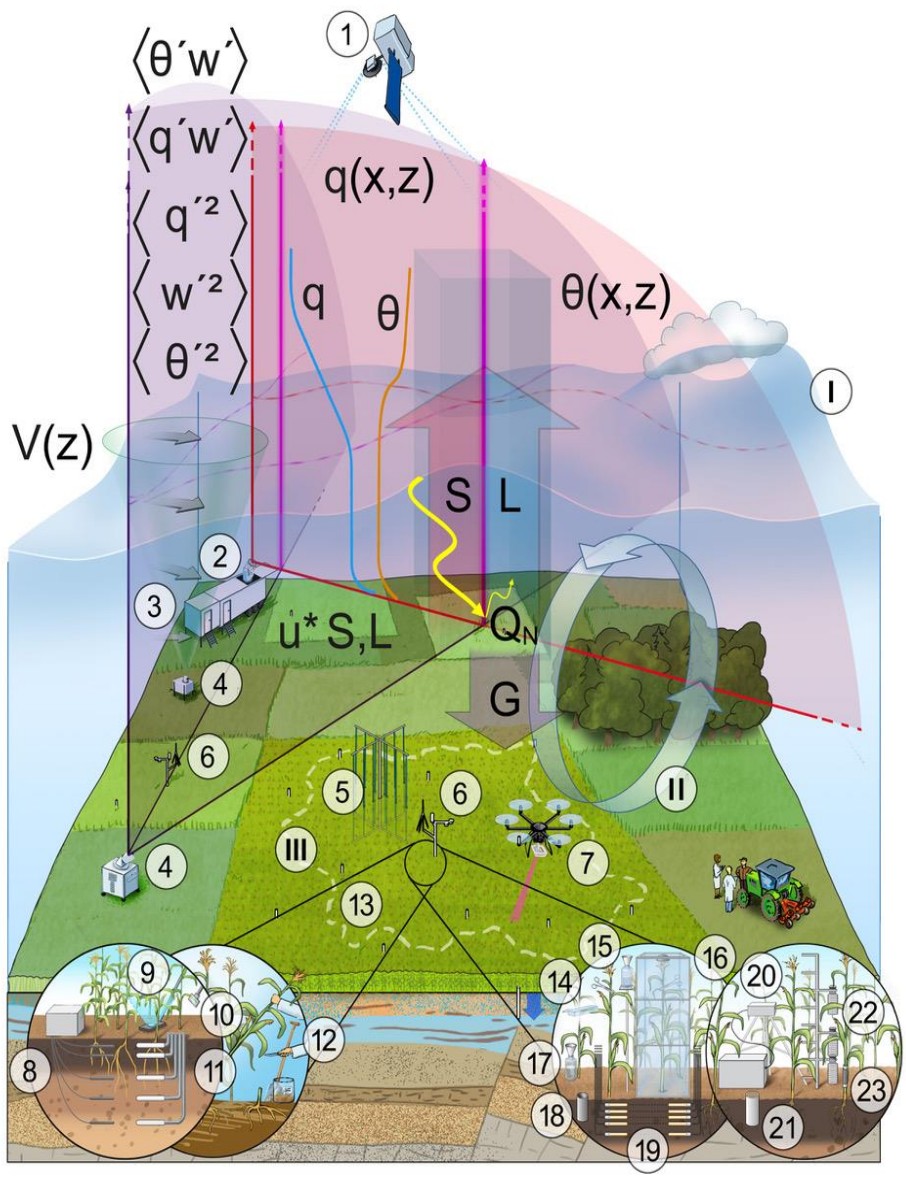

**Figure 1** LAFO sensor synergy. I: PBL top, II: mesoscale vortex, III: flux footprint. 1: Satellite remote sensing, 2, 3, 4: vertically staring and scanning Doppler, water vapor, and temperature, lidar systems, 5: 3D fiber optic distributed temperature sensing (FODS) in combination with distributed wind and trace gas sensors, 6: energy balance and eddy covariance stations, 7: unmanned aerial vehicle (UAV), 8: time-domain reflectometers (TDRs), 9: leaf area index (LAI) measurement, 10: gas exchange system for photosynthesis and transpiration rate measurements, 11: tensiometers, 12: in-situ crop measurements such as root and shoot biomass as well as canopy height, 13: soil moisture, temperature, and matric potential network, 14: leaf cuvette, 15: open rainfall sampler, 16: gas exchange chamber, 17: throughfall sampler, 18: groundwater well, 19: in-situ soil water probes (14-19 all coupled to water isotope analyzer), 20: IR gas analyzer, 21: mini lysimeter, 22: canopy temperature, water vapor, and CO2 profiles, 23: sap flow sensors. For further explanations, see www.gewex.org/panels/global-landatmosphere-system-study-panel/glass-projects and Wulfmeyer et al. (2020).


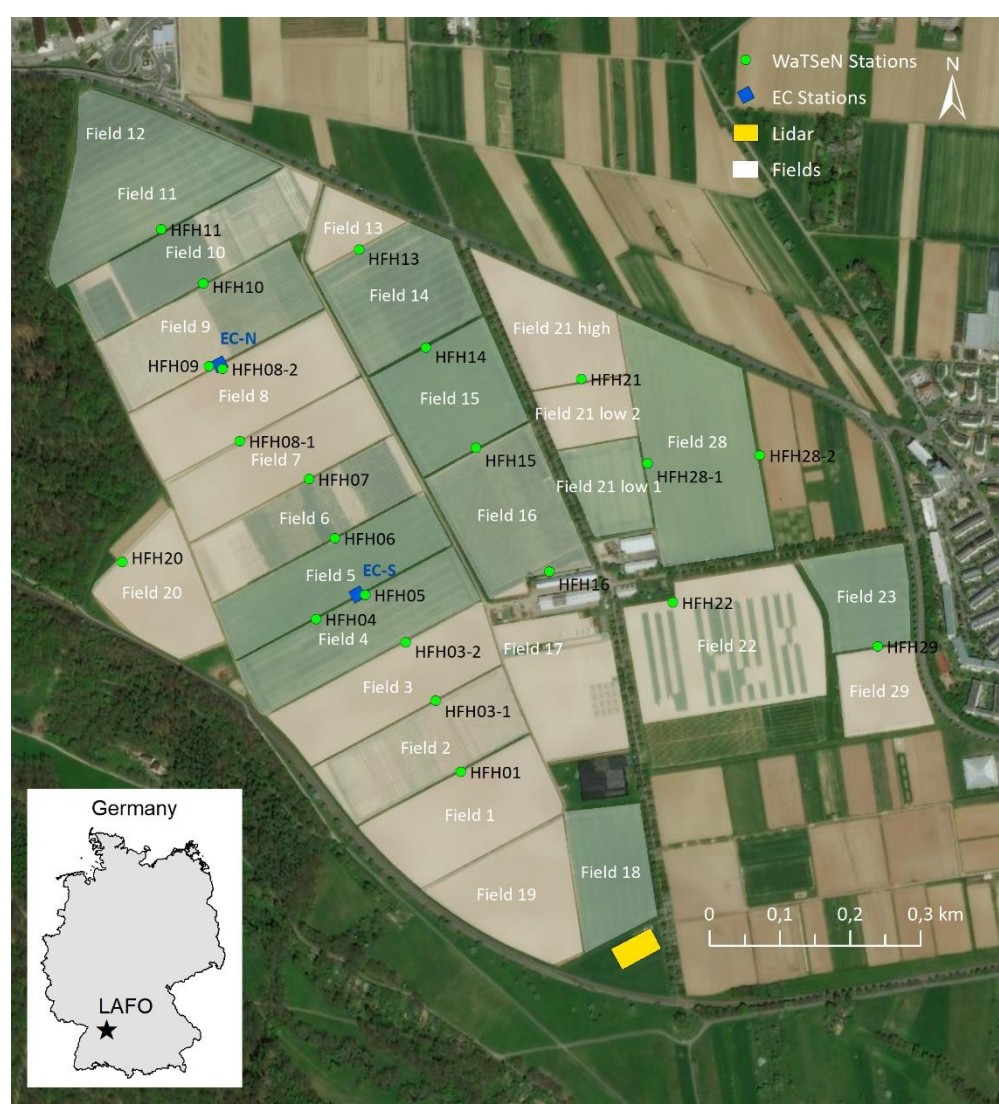

**Figure 2** LAFO experimental site at the Heidfeldhof, locations of the two eddy covariance stations, the 22 soil water content and temperature stations, remote sensing equipment at the lidar site, and extents of agricultural fields. The inlet shows the location of LAFO in Southwest Germany. Basemap source: Esri, DigitalGlobe, GeoEye, Earthstar Geographics, CNES/Airbus DS, USDA, USGS, AeroGRID, IGN, and the GIS User Community.


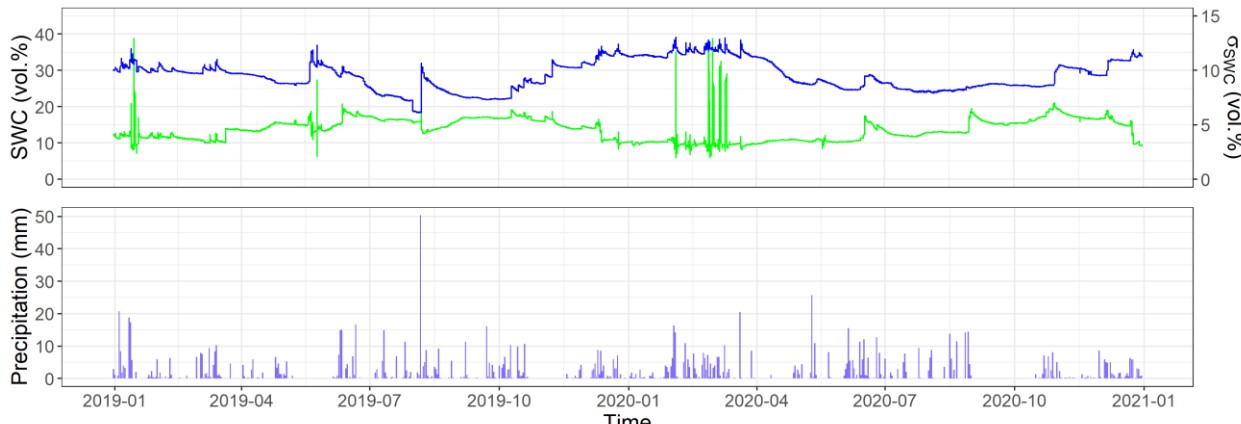

**Figure 3** Temporal variations of spatial average soil water content (upper panel, blue), standard deviation of soil water content (upper panel, green), and daily precipitation (bottom panel) with data from WaTSeN during the 2019-2020 years. Soil water content is the spatial mean of daily averaged data from 14 stations of WaTSeN.






**Figure 4** Measurements of the vertical and horizontal wind from 27 June 2021. In addition to the wind itself also the backscatter coefficient, vertical wind variance, TKE and momentum flux are shown.




**Figure 5** Water vapor mixing ratio and temperature time-series for 27 June 2021 are plotted in (A) and (B). In (C) and (D) the latent and

sensible heat fluxes are shown for that day.



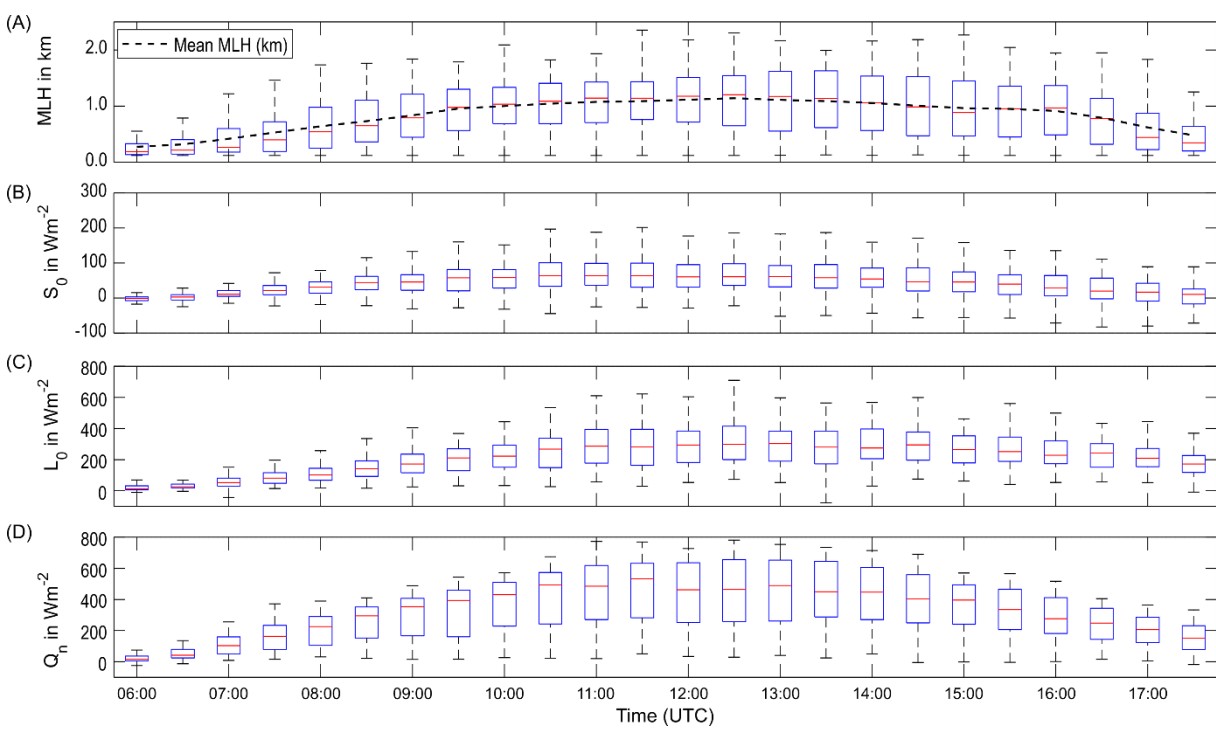

**Figure 6** Statistics of the diurnal cycle between 6 and 18 UTC of mixing layer height (A), surface sensible (B) and latent (C) heat fluxes, and net radiation (D) for three months from May to July 2021.



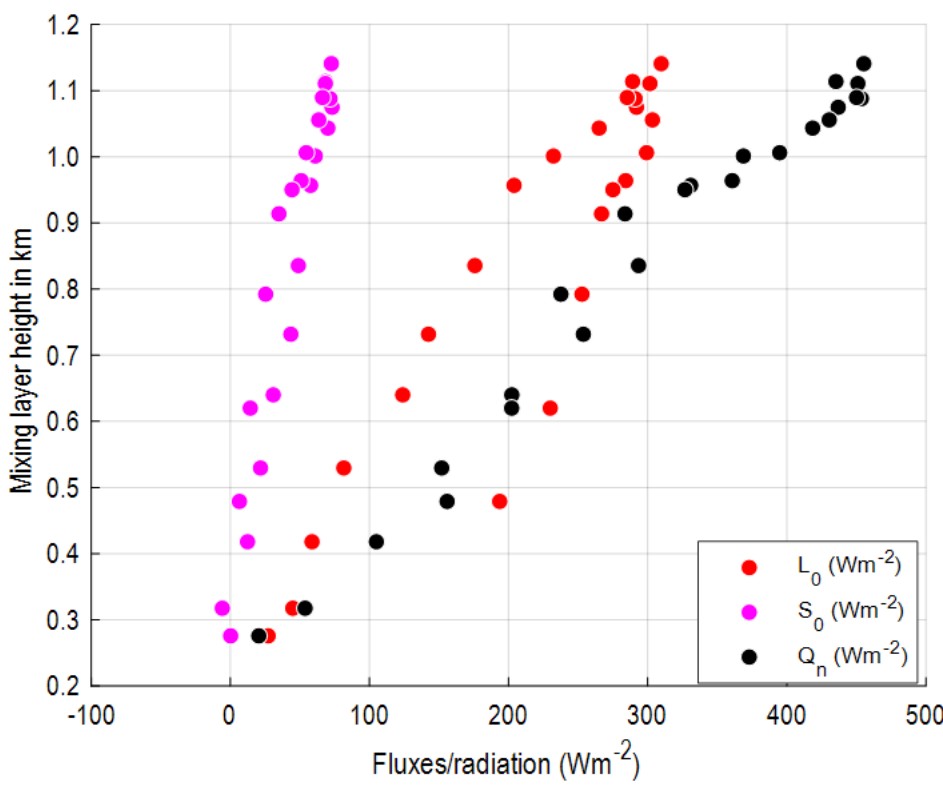


**Figure 7** Correlation of the MLH with the surface sensible and latent heat fluxes and the net radiation.

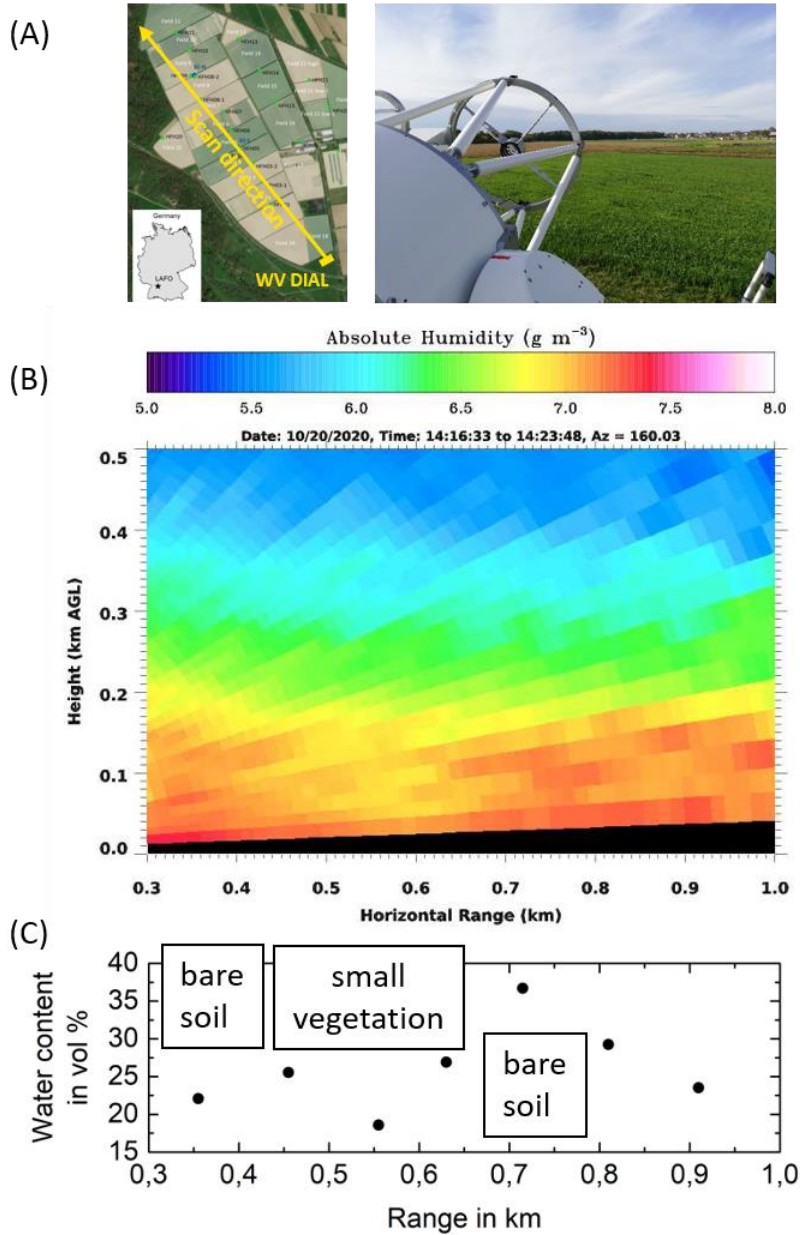

**Figure 8** Scanning WV DIAL measurements over the agricultural field of Heidfeldhof on 20 October 2020 between 14:23 and 14:31 UTC. (A) shows the scan direction to the north-west with a picture from the scanner over the agricultural fields. (B) gives the humidity field in the lowest 0.5 km above ground (with respect to the altitude of the lidar site). (C) shows the soil water content values in 40 cm depth from WaTSeN along the scan direction. Basemap source in (A): Esri, DigitalGlobe, GeoEye, Earthstar Geographics, CNES/Airbus DS, USDA, USGS, AeroGRID, IGN, and the GIS User Community.

