# Peer review of "The Land-Atmosphere Feedback Observatory: A New Observational Approach for Characterizing Land-Atmosphere Feedback"

_Geoscientific Instrumentation, Methods and Data Systems, 2022_

## Referee Comment (RC2)

Review of MS: The Land-Atmosphere Feedback Observatory: A New Observational Approach for Characterizing Land-Atmosphere Feedback by Späth et al.

**General comments**

The Authors present some sensors integrated in a new experimental site aiming to better investigate the fluxes at the land-atmosphere (L-A). The Authors describes the sensors and the way they are operated. In addition, they discuss some results, as example. Overall, I think this manuscript could be a valuable contribution to the present journal. Despite that, I think the manuscript in the present form has some limitations and it should be improved in several parts. The message should be better conveyed, the structure is in some parts redundant, English should also be corrected to some extent. Below I list my main concerns. Specific comments and minor edits are also reported that I hope can help for further improvements.

**Main concerns**

[1] as far as I have understood the added value of the new observatory is the integration of sensors to monitor ABL. And this integration can be done in any experimental sites e.g., Fluxnet, iCOS or TERENO sites. So, why not extending the existing network (L59) instead of advertising a new one with a new name? Personally, I would see more convenient (1) to show how to monitor ABL, (2) to explain the added value of these measurements, (3) to explain what to do to extend existing network to account for that. Revisiting the manuscript based on this structure would be, in my opinion, much more effective.

[2] The Authors have planned some intensive operation period (IOP). It is not clear which measurements are performed during these periods, how often these intensive periods are planned and how these activities will be operated when the project will be finished. In this sense I wonder if the operation of this new site will stop when founds are finished and the equipements running for longer term will be then similar to existing network (e.g., fluxnet etc.).

[3] Finally, the Authors argue that monitoring ABL is important to better understand L-A interaction. This is not well addressed in my opinion in the current version of the manuscript. The Authors present some results, as example, but it would help if they also show how these measurements improve our understanding of the L-A relation in comparison to more traditional sites, i.e, what would we have missed if the same site has been a more traditional fluxnet site? In addition, the discussion should be extended in my opinion considering the following questions: (1) how many sites should be equipped to have a better understanding of L-A interactions and (2) how much is feasible to extend the proposed approach to other sites? I do not expect that the Authors can fully address these questions, but I believe these are important issues to integrate in the discussion that could also strengthen the value of the proposed approach.

**Specific comments**

L11-L16. There are three components. Atmosphere is related to lidar; soil and land-surface with eddy covariance stations and soil moisture and temperature network; what is performed for vegetation is not indicated in the abstract and should be integrated.

L22. I think the term "important" is too vague and the Authors should better show and explain what exactly the added value of these measurements is. See also general comment #3 above.

L56-57. not clear. Something wrong with the English grammar?

L59. Following Helbig et al, the question is how to extend existing network and it is not clear to me why to argue for a new experimental site with a new name (see general comment #1 above).

L111-L112. It sounds strange to have same titles. Please consider rewording, e.g., 2. Observation strategy 2.1. Targeted variables

L131-L145. In this text there is a mix between the description of the targeted variables and how they will be measured. But this is redundant as it should better fit into section 2.3. Please consider moving descriptions.

L140. What are agricultural measurements? What are soil probes?

L145-173. Objectives sounds better for me in the introduction part. In addition, these objectives should be, at least partially, addressed by the results and the analyses presented and discussed. This is well presented in the manuscript because these objectives are not reported in the discussion (see also general comment #3).

L269. Why field capacity and not higher values e.g., saturation? Please also note that field capacity is a debated parameter. I encourage to be more precise.

L302. How is LAI measured/estimated?

L369. "does not decrease" instead of "decreases not"

L399. What is measured for vegetation should be listed (same comment as L11-L16).

L392. In the conclusion section I would extend the acronyms for easy of interpretation (LAFO, GLAFO.

L414. Out? Database.

L414. How the connection is implemented could also be described.

Table 1 can be improved specifying temporal and spatial resolution, and extended to measurements conducted during the intensive operation period (see general comment #2)

Figure 1 shows much more instruments than what is actually described within the manuscript e.g., tensiometer, lysimeter etc. The description should be integrated in the main text.

Figure caption and legend should be more self-explained. E.g., figure 7, explain MLK, L0, S0, Qn

---

## Author Comment (AC1)

**Späth et al., 2022, The Land-Atmosphere Feedback Observatory: A New Observational Approach for Characterizing Land-Atmosphere Feedback**

Dear Editor,
Thank you for your consideration of our manuscript. We will answer the reviewers' comments point by point. We will indicate any changes using line number for the revised manuscript and send a track-changes version with markup.

**Reviewer #1: Manuel Helbig**

The present study describes the setup and goals of the Land-Atmosphere Feedback Observatory (LAFO). LAFO combines atmospheric, vegetation, and hydrological measurements to address pressing questions on land-atmosphere feedbacks. The authors give a broad overview of the different components and how they are integrated in LAFO. In my opinion, the manuscript represents an important contribution to the literature outlining how to design an observatory that addresses multi-disciplinary questions at the interface of atmosphere and land.

Manuel Helbig, we thank you very much for taking the time to review our manuscript and to provide expert insights and suggestions. We would like to answer your comments point by point.

Please see below some minor comments:

Line 78: Explain why these lidars were important.
    Scanning lidars are key, especially for humidity and temperature measurements, to overcome the blind near range of vertical point lidars due to no full overlap of outgoing laser beam and field-of-view of the receiving telescope. Only with scanning lidar, observations close to the surface are possible.
We modified text as:
Operating scanning lidars were the key to obtain measurements in the surface layer region to overcome the near-range gap of vertical pointing lidars which deliver no data of the surface layer (Späth et al., 2022). (Line 81).

Line 82: Do the authors target specific parameterizations in specific models? If yes, then it should be described here in more detail.
    In general, we do not want to limit the investigation of parameterizations. We are open for all kinds of studies for which our data are suitable. We are willing to share our collected data freely with other colleagues.
For instance, currently, we are dealing with the applicability of the Monin-Obukhov similarity theory (MOST) in natural heterogeneous terrain.
We added MOST as example in this paragraph (Line 88) which is mentioned a few lines below as example (Line 109).

Line 84-87: Are there more LAFOs planned? How will this be achieved?
    Yes, LAFO can be considered as a prototype observatory for the GEWEX LAFOs (GLAFOs). As further GLAFO sites, we propose locations where already soil and surface measurements exist (such AmeriFlux sites) or sites that are well equipped with remote sensing systems for studying the PBL.

Currently, the Ruisdale Observatory in The Netherlands and the MOL-RAO of DWD in Germany agreed to join the GLAFO initiative. Further proposed sites are found in the GLAFO White Paper (see https://www.gewex.org/panels/global-landatmosphere-system-study-panel/glass-projects).

Line 88: Soil measurements?

Thanks for that request. Multi-level soil measurements are included in the energy balance/eddy-covariance stations but we added them now separately for clarity. We modified the text accordingly. (line 93)

Line 93: What is the spatial and temporal scale of the targeted feedbacks?

Our study region are the research fields of Heidfeldhof which covers an area of 0.84 km$^2$. In addition, we have a forest and urban areas close to our study region which can also be target of interest and be reached by our remote sensing instruments. With IOPs we can focus on certain case studies but with the operational obtained dataset we are able to extend studies to different daytimes, seasons or weather conditions.

Line 119: Which instruments are used to measure these variables? What is the depth/extent of these measurements?

According to reviewer #2 we separate the target variable from the description how we measure these variables. The instrumental description is given in Sect. 3, particularly for the atmospheric variables in Sect. 3.1. We added some information about the measurements (up to 10 km), however, our focus region is the ABL. For more information and details about the instrument performance we cite the relevant references.

Line 121: How is the boundary layer height derived? Based on which measurement?

We derive the boundary layer height from the lidar measurements. Depending on the kid of lidar we use, DIAL, RRL or DLs, we can determine boundary layer height by the largest gradient of humidity, temperature or backscatter signal. Thus, we achieve the same spatial and temporal resolution as for the key variables.

Line 130: When is most used and when is EC used?

It depends on availability of observations whether lidar data od EC data are used to combine with MOST. We rewrote the sentence to be more clear.
"…are determined by combining the scanning lidars with MOST (Späth et al., 2022) or EC station measurements with MOST (Wulfmeyer et al., 2022b, accepted and in press). (Line 144)

Line 136: What is the spatial (and temporal) resolution of WaTSen?

As depicted in Fig. 2, WaTSeN is distributed over the research farm Heidfeldhof covering an area of 0.84 km$^2$. The temporal resolution is 15 minutes. We added this information to the manuscript. (Line 155)

Line 137: How can evaporation be determined? How is drainage accounted for?

Evapotranspiration can be determined using the eddy-covariance measurements. It cannot be determined from the WaTSen network alone, yet. For this, additional instrumentation is required, most importantly net radiation. Drainage can only be inferred as a residual term based on the change in soil water store and the surface fluxes, providing the latter are quantifiable. (see Line 157)

Line 164: Unclear what this means. Please clarify.

We agree. This was a little unclearly and shortly stated.

In addition, the LAFO site in situ measurements facilitate straightforward comparison between measured and modelled time series of evapotranspiration, soil water content, soil temperature, soil matric potential, plant growth, plant development, leaf area index, yields, and the energy balance closure at the surface, all of which are considered important state variables to further develop the agroecological multi-model library Expert-N, and the land surface model NOAH-MP-GECROS. This further development is a prerequisite for coupling spatially explicit land surface models with large eddy simulations to account for field-to-field variability in crop/vegetation types and dynamics. (Line 181)

Line 196: How often is LAI measured?

LAI measurements are taken every second week during the main vegetation period together with the BBCH code to capture the vegetation development. We added this information to our manuscript in Sect. 3.3 (Line 339).

Line 239 and elsewhere: It is not always clear why some of the atmospheric instruments are not continuously deployed. Please explain why.

Particularly, DIAL and RRL are research instruments which are not automated and still undergo further development to advance their capability and performance. Thus, these two instruments are operated manually during dry periods. For scanning operation reasons of eye-safety also need to be accounted for which we do not have an automated safety procedure yet. ARTHUS is an automated and remote-controllable system, however, recently it was involved in other projects and field experiments and, thus, not continuously located and operated in LAFO. Otherwise all other described instruments are operated continuously.
We added more information about our research instruments in line 256 and we marked the continuously operated instruments in Table 1 in bold face for more clarity.

As research instruments, the scanning DIAL and RRL are not automated, thus, the operation requires human effort and are only operated during dedicated intensive observation periods for certain time periods. Whereas ARTHUS is a full automated system to collect vertical profiles. When ARTHUS is not involved in other field campaigns, it is operated in LAFO. (Line 256)

Line 274: A simple correlation approach might not be sufficient. Could a more complex model like Penman-Monteith be used to explore soil moisture controls?

You are probably right, that we followed here a simple correlation approach which was sufficient for what we observed so far. For a more sophisticated analysis of soil moisture controls we suggest to include the energy balance by using net radiometers (especially with different vegetation at the site).

Section 3.3: I find that this section is not well integrated. How are all these measurements used to better understand land-atmosphere feedbacks? This should be explained here.

It is our intention to give here the description of how to measure the variable in LAFO. The variable description is given in Sect. 2.1.

Line 362: How much lower is the MLH?

For cloudy days the MLH is about 400 m lower. We added this information to our manuscript (Line 400)

Line 363: Are these daily averages? Please clarify? Reporting daily averages would be meaningful.

The fluxes and MLH are for each 30-minute period averaged over the 3-month period. So the statistics in of Fig. 6 reporting the daily averaged values.

Data of 1s MLH data are averaged for 30-minute intervals as the surface fluxes are calculated likewise for half-hour timeslots. (Line 389)

Line 369: This is not surprising. Sensible heat flux should be the main driver of ML growth.

You're right, this is no surprise. ML growth is driven by sensible heat flux which we confirm here for our measurements.

Table 1: I would recommend to avoid as much as possible abbreviations in this table. Also, more information on what these variables can be used for would be helpful here.

Thanks for your recommendation. We removed as much as possible abbreviations in Tab. 1 or give their explanation in the caption. Otherwise variable names are explained in the text.

Review of MS: The Land-Atmosphere Feedback Observatory: A New Observational Approach for Characterizing Land-Atmosphere Feedback by Späth et al.

**General comments**

The Authors present some sensors integrated in a new experimental site aiming to better investigate the fluxes at the land-atmosphere (L-A). The Authors describes the sensors and the way they are operated. In addition, they discuss some results, as example. Overall, I think this manuscript could be a valuable contribution to the present journal. Despite that, I think the manuscript in the present form has some limitations and it should be improved in several parts. The message should be better conveyed, the structure is in some parts redundant, English should also be corrected to some extent. Below I list my main concerns. Specific comments and minor edits are also reported that I hope can help for further improvements.

We thank the anonymous reviewer for her/his critical reading of our manuscript to improve before publication. With our LAFO we do not only be a new site for flux measurements at the land-surface but want to observe the full extent of the land-atmosphere system including soil, land-surface, vegetation and lower atmosphere. Having variables observed simultaneously in all of these compartments will allow to investigate relations and feedbacks between variables.
Below, please find our responses point-by-point.

**Main concerns**

[1] as far as I have understood the added value of the new observatory is the integration of sensors to monitor ABL. And this integration can be done in any experimental sites e.g., Fluxnet, iCOS or TERENO sites. So, why not extending the existing network (L59) instead of advertising a new one with a new name? Personally, I would see more convenient (1) to show how to monitor ABL, (2) to explain the added value of these measurements, (3) to explain what to do to extend existing network to account for that. Revisiting the manuscript based on this structure would be, in my opinion, much more effective.

For this purpose, we developed a hierarchical design matrix for the GLAFOs (see GLAFO White Paper) and considered the existing networks. However, there are also key observatories such as MOL-RAO that are very well equipped to study L-A feedback. Also, measurement sites are needed in regions where even no networks exist such as in Africa. In any case, in addition to the observations of the soil, land-surface and vegetation we deployed a special set of instruments to monitor the ABL. Special for LAFO and this is why a new observatory makes sense are the unique scanning lidars.
Our idea to study L-A interaction started 2014 with the field experiment SABLE (Surface-Atmosphere-Boundary-Layer Exchange) within a research unit when we realized that observations of ABL are beneficial. Since then we developed our measurement strategy which was realized first within LAFE and now elaborated in LAFO.
The key instruments for our sensor synergy are the scanning lidars particular the unique scanning DIAL and RRL for humidity and temperature measurements in the surface layer and throughout the ABL. For that we do not get only profiles but also 2D structures. DIAL and RRL are research instruments which are going to be further developed and improved. Consequently, to combine both, L-A feedback investigations and lidar development, we decided to setup LAFO as new observatory close to our university. In addition, as our study region is used for agricultural research by our agricultural faculty, we can make use of previously realized investigations of the soil and plant conditions in this area.

As recommended by Helbig et al. (2021) the expansion of existing sites of FLUXNET, ICOS and TERENO would be helpful to capture different weather and climate situations. However, we think in these networks automated instruments are needed which is not yet available for scanning DIAL and RRL.

[2] The Authors have planned some intensive operation period (IOP). It is not clear which measurements are performed during these periods, how often these intensive periods are planned and how these activities will be operated when the project will be finished. In this sense I wonder if the operation of this new site will stop when founds are finished and the equipements running for longer term will be then similar to existing network (e.g., fluxnet etc.).

The LAFO has the status of a research facility of the University of Hohenheim. Therefore, continuity of the measurements and the required support is ensured.
As observatory we seek for long-term observations and continuous dataset. The setup phase of LAFO was project funded, however, meanwhile LAFO has been established as official research facility of University of Hohenheim and will be continued. IOPs are implemented mainly during the vegetation period in spring and summer time. For IOPs also external partner are welcome to setup additional instrumentation to complement our observations, e.g. fiber-optic temperature sensing, which will be project-based.

[3] Finally, the Authors argue that monitoring ABL is important to better understand L-A interaction. This is not well addressed in my opinion in the current version of the manuscript. The Authors present some results, as example, but it would help if they also show how these measurements improve our understanding of the L-A relation in comparison to more traditional sites, i.e, what would we have missed if the same site has been a more traditional fluxnet site?

In our manuscript we show three measurement examples to illustrate our measurement capability of LAFO instrumentation. It is not the intention here to include a full analysis at this point. We added the targeted objective to each of the examples to guide the reader more clearly.

In addition, the discussion should be extended in my opinion considering the following questions:
(1) how many sites should be equipped to have a better understanding of L-A interactions and

There is a GLAFO white paper of GLASS panel members, which suggests to realize at least one GLAFO in all climate zones. The white paper (see comment of reviewer 1) proposes to extend, e.g., network sites of FLUXNET, ICOS or TERENO in different levels of instrumentation according to the location and funding possibilities.

(2) how much is feasible to extend the proposed approach to other sites? I do not expect that the Authors can fully address these questions, but I believe these are important issues to integrate in the discussion that could also strengthen the value of the proposed approach.

As explained in the GLAFO White Paper, the equipment of GLAFOs sites will be organized along specific levels of complexity according to a hierarchical design matrix. The highest complexity but also the best output is achieved with our scanning DIAL and RRL that need to be developed and operated also at other sites. As recommended by Helbig et al. (2021), any profiling instruments to monitor the ABL would be already beneficial. Most important is the extension of the observational equipment with a thermodynamic profiler (e.g., water-vapor and temperature Raman lidar) because these profiles are essential for the derivation of L-A feedback metrics focusing on the atmospheric leg. Doppler lidars can also contribute with measurements of horizontal wind profiles and vertical velocity statistics This will allow studies like we presented in Sect. 4.2. Our scanning DIAL and RRL maybe moved to other GLAFOs for dedicated IOPs similar when these instruments were moved to the ARM SGP site for LAFE 2017.

**Specific comments**

L11-L16. There are three components. Atmosphere is related to lidar; soil and land-surface with eddy covariance stations and soil moisture and temperature network; what is performed for vegetation is not indicated in the abstract and should be integrated.

Thanks for pointing this out. We added the vegetation measurements in the abstract (Line 16).

L22. I think the term "important" is too vague and the Authors should better show and explain what exactly the added value of these measurements is. See also general comment #3 above.

We rephrased this sentence.
Both IOPs and long-term observations will provide new insight into exchange processes and their statistics for improving the representation of L-A feedbacks in climate and numerical weather prediction models. Especially the lidar component will support the investigation of coupling to the atmosphere. (Line 23).

L56-57. not clear. Something wrong with the English grammar?

We corrected this sentence to: "L-A interactions at the surface are studied globally based on observation using eddy covariance (EC) stations." (Line 60)

L59. Following Helbig et al, the question is how to extend existing network and it is not clear to me why to argue for a new experimental site with a new name (see general comment #1 above).

New GLAFOs are essential in regions where other observatories do not exist yet. Otherwise we agree with Helbig et al. (2021) and considered in the GLAFO White Paper already to extend stations of networks with instruments for atmospheric measurements. This means automated systems for wind, temperature and humidity observations in the ABL which are already commercially available. The uniqueness of LAFO are the scanning capability of the research lidars for temperature and humidity measurements. This advances LAFO with respect to commercially available instruments (see also our response above).

L111-L112. It sounds strange to have same titles. Please consider rewording, e.g., 2. Observation strategy 2.1. Targeted variables

Thanks for this comment. We followed your suggestion and renamed Sect. 2.1: Targeted Variables (Line 133).

L131-L145. In this text there is a mix between the description of the targeted variables and how they will be measured. But this is redundant as it should better fit into section 2.3. Please consider moving descriptions.

Thanks for your careful reading. We rephrased this paragraph in Sect. 2.1 to the description of the targeted variables and left the description of how to measure to Sect. 3.3.

To study vegetation status and their spatial pattern in standing crops we will record information about biomass, plant height $z_P$, plant density $\rho_P$, plant humidity content $q_P$, skin temperature $T_S$ of bare soil, normalized difference vegetation index (NDVI), leaf area index (LAI) and nitrogen supply (red edge inflection point, REIP). The vegetation development status is registered according to the BBCH scale (Biologische Bundesanstalt für Land- und Forstwirtschaft, Bundessortenamt und Chemische Industrie; Meier, 2018). (Line 146)

L140. What are agricultural measurements? What are soil probes?

As cited, Schutte (2005) studied traction measurements during tillage to identify differences in soil such as clay content variations. Thanks for the hint. Here we meant soil samples and modified the text accordingly. (Line 159)

L145-173. Objectives sounds better for me in the introduction part. In addition, these objectives should be, at least partially, addressed by the results and the analyses presented and discussed. This is well presented in the manuscript because these objectives are not reported in the discussion (see also general comment #3).

We agree and the LAFO objectives are given in the introduction (see Line 96). At this point in the manuscript we describe how we are going to target the objectives.

L269. Why field capacity and not higher values e.g., saturation? Please also note that field capacity is a debated parameter. I encourage to be more precise.

We somewhat agree. It is true that field capacity, defined at a certain pressure head, is not always a useful concept to quantitatively evaluate field measurements. This was not our intention, anyway. To remove ambiguity and qualitatively contextualize the results, we now write:

The SWC shows the typical wetting and drying cycles during the year reaching field capacity (the water content which can be held by capillary forces against gravity) in winter and subsequently long periods of drying between April and October as a result of net evapotranspiration being larger than precipitation over this period. SWC above field capacity are only observable during strong infiltration events and subsequent rapid drainage leads to plateaus in the observations, which is testimony to the high (un)saturated soil hydraulic conductivity. (Line 299)

L302. How is LAI measured/estimated?

We measure LAI with LAI2200C sensor from LI-COR Biosciences Inc. (USA). We added this information to our manuscript. (Line 336)

L369. "does not decrease" instead of "decreases not"

Manuscript changed accordingly (Line 407)

L399. What is measured for vegetation should be listed (same comment as L11-L16).

We included the vegetational measurements in this section (Line 444)

L392. In the conclusion section I would extend the acronyms for easy of interpretation (LAFO, GLAFO.

We followed your suggestion and modified our manuscript accordingly. (Line 436)

L414. Out? Database.

This should read "Our database…" and was corrected in the manuscript. (Line 462)

L414. How the connection is implemented could also be described.

Thanks for this comment, good idea. Our database bases on the data portal software of TEODOOR (Kunkel et al., 2013) which is open-source available. Indeed, might be interesting for others, thus, we added this information to our manuscript and cite the corresponding reference giving more details. (Line 462)

Table 1 can be improved specifying temporal and spatial resolution, and extended to measurements conducted during the intensive operation period (see general comment #2)

Thanks for the comment. We added the temporal and spatial resolutions for the continuously operated instruments which run with fixed settings. For the instruments deployed during IOPs we refer to the instrument description in Sect. 3 and for even more details to the cited references. We think we cannot give one setting for all conditions of IOPs for which the instruments are operated. We also do not claim completeness of instruments to be involved in future IOPs.

We revised Table 1 accordingly.

Figure 1 shows much more instruments than what is actually described within the manuscript e.g., tensiometer, lysimeter etc. The description should be integrated in the main text. Figure caption and legend should be more self-explained. E.g., figure 7, explain MLK, L0, S0, Qn

We thank you for that comment. We added more information about the extended instrumentation in the text as recommended (Line 119). Further, we added more information to the figure captions where needed.